# The Experiences of Living with a Visual Impairment in Peru: Personal, Medical, and Educational Perspectives

**DOI:** 10.3390/ijerph22070984

**Published:** 2025-06-23

**Authors:** Jorge Luis Cueva-Vargas, Claire Laballestrier, Joseph Paul Nemargut

**Affiliations:** 1Programa de Investigacion Formativa, Universidad Cesar Vallejo, Trujillo 13001, Peru; 2École D’optométrie, Université de Montréal, Montréal, QC H3T 1P1, Canada; claire.laballestrier@umontreal.ca; 3Centre de Recherche Interdisciplinaire en Réadaptation (CRIR) du Montréal Métropolitain, Montréal, QC H4B 1P3, Canada; 4Centre Institut Nazareth et Louis-Braille, CISSS Montérégie-Centre, Longueuil, QC J4K 5G4, Canada; 5Centre de Réadaptation Lethbridge-Layton-Mackay, CIUSSS du Centre-Ouest de l’Île de Montréal, Montreal, QC H4B 1P3, Canada

**Keywords:** visual disability, assistive devices, vision rehabilitation, eye care, accessibility

## Abstract

Background: Nearly 5 million people in Peru live with visual impairments, many of which are irreversible. within addition to eye care services, these individuals could benefit from government services and rehabilitation to improve their quality of life and promote equitable, inclusive social participation. Although numerous government policies address this, little is known about their perception and implementation. Methods: Semi-structured individual online interviews were conducted with 29 people (7 low vision, 12 blind, 6 educators/rehabilitators, 4 medical doctors) in Peru between July and November 2024. Each participant was asked to respond to the same 16 open-ended questions. Their transcripts were coded into themes in 5 domains: assistive devices, vision rehabilitation services, government assistance programs, accessibility for people with visual impairments, and eye care services. The themes were compared among members of each group. Results: Themes from educators/rehabilitators aligned well with those with blindness but much less with ophthalmologists and those with low vision. Participants mentioned that assistive devices are not traditionally provided by the government. There was little mention of vision rehabilitation services, particularly from low vision participants. Additionally, participants with visual impairments mentioned a lack of sensitivity from teachers, employers, and transport drivers. Interestingly, none of the participants with visual impairments benefitted from financial assistance. Conclusions: Many of the barriers are societal, referring to the lack of understanding from the public in relation to employment, education, transportation, or the use of assistive devices. People with visual impairments and educators should be included in any policy decisions to promote equality for Peruvians with vision loss.

## 1. Introduction

Almost 21 million people in Andean Latin America—which includes Venezuela, Colombia, Ecuador, Peru, Bolivia, and Chile—are living with a visual impairment, and more than 870,000 of these individuals are blind [1]. Peru is the second Andean country, after Colombia, with the highest visual impairment (VI) prevalence, with 13.5% of its population affected at all levels, from mild VI to blindness, including near vision problems as reported in 2020 [1,2,3]. The most common eye diseases prioritized by the Peruvian government include retinopathy of prematurity, refractive errors, cataracts, glaucoma, and diabetic retinopathy [2,4]. In 2017, VI was the most prevalent functional disability in Peru, affecting 48.3% of individuals with disabilities, a higher rate than any other functional disability [5].

Peru is among 184 out of 193 United Nations (UN) member states to ratify the Convention on the Rights of Persons with Disabilities (CRPD), implementing it in 2008. The CRPD states that independent living must be facilitated by the elimination of accessibility barriers, whether they are found in the physical environment (i.e., schools, workplaces, indoor or outdoor facilities, roads, health infrastructures), transportation, or information and communications. In that regard, it emphasizes that these facilitators should include braille and that technologies should be accessible at a minimum cost [6]. By ratifying the CRPD, Peru legally adopts anti-discrimination laws and commits to eliminating accessibility barriers [7]. Furthermore, the adoption of the CRPD’s Optional Protocol enables individuals to submit direct complaints to the UN Committee on the Rights of Persons with Disabilities [7,8].

Peru has established a comprehensive legal framework to protect the rights of persons with disabilities with implications for vision rehabilitation services. Indeed, the 1993 Constitution safeguards disability rights through Articles 7 and 23 (healthcare, social protection, and employment), as well as Article 16 (inclusive education). Peru also emphasizes independent living with community inclusion support (Article 9), personal mobility (Article 20), inclusive and lifelong education (Article 24), and access to essential services, including local rehabilitation programs and comprehensive habilitation services, all of which are particularly relevant for people with visual impairments (PVI) (Article 25 and 26). Article 27 reinforces workplace inclusion by prohibiting discrimination and mandating reasonable accommodations. The legal frameworks emphasize the specific needs, such as vision rehabilitation and orientation and mobility (O&M) training, as critical to fostering independence and full participation in society [9]. These commitments are further reinforced by national legislation, notably the General Law on Persons with Disabilities (Law No. 29973, 2012) [10] and its implementing regulation, Supreme Decree No. 002-2014-MIMP [11].

Vision rehabilitation is a range of services aiming to maintain, improve, or restore the quality of life and the independence, prevent accidents and falls, and improve or help to maintain the physical and psychological well-being of PVI [12]. Action Plan 2014–2019 on eye health included vision rehabilitation as an essential part of a person-centered healthcare [13]. These services should be accessible to any person with a visual impairment, regardless of age and whether the person lives in a rural or in an urban area, and provided by a sufficient number of trained professionals, ideally with the government subsidizing them to ensure equity [14]. As for eye care services, they should meet the same criteria as the vision rehabilitation services. If free services are impossible, they should be affordable or integrated in a universal health coverage plan and be accessible without delay for emergencies or within a reasonable time frame for a visual assessment [12]. Eye care services are crucial to collecting data on the prevalence of VI to prevent avoidable blindness, to correct refractive errors causing mild to severe VI, to treat curable diseases, or even to monitor degenerative and lifelong diseases [12]. Both eye care and vision rehabilitation services should be integrated into general health systems [15].

Unfortunately, the research available on eye care services in Peru is scarce, even when looking at broader research over Latin America or globally, whereas vision rehabilitation studies are practically nonexistent. A Hong et al.’ (2016) study on universal eye health in Latin America states that Peru has a ratio of 3.13 ophthalmologists per 100,000 inhabitants, which places it below the regional average of 5.2 per 100,000 inhabitants. Moreover, according to the same study, Peru is the third most unequal Latin American country relating to the regional repartition of ophthalmologists [16], who tend to settle in the more densely populated regions and cities because of better salaries and career opportunities they offer [17,18]. The biggest and financially most attractive Peruvian cities are found along the coast, while the mountains and the jungle, poorer and less densely populated, have diminished access to healthcare in general and, thus, to eye care services [19]. Lima alone is home to 38% of the Peruvian population with a VI [5]. In their study about the determinants of eye care service utilization in Peru, Barrenechea-Pulache et al. (2022) [19] found that the population who used these services the most was generally wealthier and more educated than those who did not. Indeed, people with a lower education level tended to be unaware of possible treatments for their diseases or even reluctant to use them. The authors also noted that if people benefiting from a health insurance (i.e., ESSALUD) were more inclined to use the eye care services than the ones who were unprotected, more than 50% of the insured did not access them because of various accessibility difficulties [19].

Little is known about the state of vision rehabilitation in Peru. According to Chiang et al. (2011), Peru has a low vision service coverage, between 11% and 50% [14]. Between 50 and 99 primary research studies have been conducted on vision and eye health in Peru between 2000 and 2019, which places the country in the second-to-last category [12]. Such a mapping of the state of vision rehabilitation research is not available. As for vision rehabilitation services, Chiang et al. (2011) state that, as for the prevalence of VI and blindness, the difficulties in understanding the increasing need are due to insufficient systematic data [14]. Taking into account the person-centered approach supported by the World Health Organization (WHO), vision rehabilitation services should be integrated directly into the eye health services and be more accessible to the people who need them. It is also crucial that PVI receive a continuum of services, including help for cultural, as well as social, participation [12]. Scientific literature is generally lacking on the state of eye health in Peru, particularly vision rehabilitation. The studies that do exist on this last aspect do not provide precise knowledge of the situation in Peru itself because they use a survey administered to experts, which certainly represents the clinical perspective of the visual impairments, but is not person-centered. The present study explored the perspectives and the day-to-day reality of Peruvians with a visual impairment, as recommended by the World Report on Vision [15]; it also offers the perspective of educational, rehabilitation, and medical professionals with the goal of comparing their insights and perspectives, intended to be as representative as possible. It also addresses the first and the second objectives of WHO’s Global Action Plan 2014–2019 on eye health. This study aims to generate evidence of the magnitude of eye care services and their effectiveness, exploring in the interview guide the barriers faced in accessing healthcare, rehabilitation, support and assistance, their environments, education, and employment. To address this aim, the study explores the following research questions: (a) What are the perceptions of vision rehabilitation services received in Peru by people with visual impairments? (b) What are the perceptions of accessibility for people with a visual impairment regarding barriers associated with education, independent mobility, and activities of daily living? and (c) What are the perceptions regarding government assistance programs provided to people with visual impairments?

The overall goal of the study was to develop professional recommendations to improve the social participation and well-being of PVI based on the input from medical, educational, and rehabilitation professionals, as well as people with a visual impairment in Peru.

## 2. Materials and Methods

### 2.1. Recruitment and Sampling

All participants were recruited through personal and professional contacts in the area between July and November 2024. In addition, anyone who wished to distribute the invitation to their friends, colleagues, or acquaintances who were eligible was encouraged to do so, thus, creating a “snowball effect” that expands the sample and allows for the recruitment of more participants [20]. Purposive sampling was performed to recruit and select a diversity of participants in different areas of the country to collect information from a wider area [21]. Recruitment continued until theoretical data saturation was obtained for each group of participants.

### 2.2. Eligibility Criteria and Ethical Considerations

The participants were categorized according to their professional status and personal experience. For PVI, they were required to be at least 18 years old and self-identify as having an irreversible VI that was diagnosed at least one year ago and that causes an incapacity in their ability to perform activities of daily living. Participants who were professionals working with PVI (in the areas of medicine, education, or rehabilitation) were required to self-identify as currently working, having worked in the field for at least one year, and having a contact with PVI at least monthly. Professionals with visual impairments were interviewed based on their professional expertise and not their personal experience during the interviews. The study was conducted in accordance with the Declaration of Helsinki. The ethics approval was obtained from the Comité d’Éthique de la Recherche Clinique (CERC) of the Université de Montréal (# 2024-5924).

### 2.3. Procedure

To better understand the perceptions relating to the realities of PVI, a qualitative research study using in-depth semi-structured interviews was conducted. Members of the community living with visual impairments, medical and rehabilitation practitioners, as well as educational staff for students with visual impairments, were interviewed by authors J.LC.V. or C.L. on questions related to each of the following subjects: assistive devices, vision rehabilitation services, government assistance programs, accessibility for PVI, and eye care services (see Appendix A for additional information). These subjects correspond to the gaps and barriers to be overcome identified by the latest Action Plan of the WHO [13] and previous research [14]. Participants were requested to respond to each of the questions to the best of their current knowledge. There was no time limit placed on answers to any of the questions. The interview was recorded on the Zoom platform for later analysis.

### 2.4. Analysis

Interviews were conducted privately by the authors, with each session lasting between 30 and 60 min. Upon completion, the recordings were securely stored on a password-protected drive. The interview data were transcribed verbatim from the recorded audio files using Descript, an AI-powered transcription application. The transcripts were then uploaded for further processing and MAXQDA Analytics Pro 24 [22] for data analysis.

To facilitate data management, the following categories were predetermined based on the interview questions: assistive devices, vision rehabilitation services, government assistance programs, accessibility for PVI, and eye care services. Inductive thematic analysis was conducted to determine the themes based on the verbatim codes in the transcripts. For instance, the category “assistive devices” was predetermined , but the themes “payment structure”, “access”, and “devices” emerged directly from the data.

Data coding was performed by J.L.C.V., with the entire process and findings reviewed collaboratively by all authors to ensure accuracy and consistency. Each theme was derived from sets of codes based on the transcripts of the recorded audio files. The authors reviewed the transcripts and recordings and discussed the themes on a biweekly basis to ensure they accurately reflected the stated views and opinions of the participant. Errors were corrected manually when the transcript did not accurately match the audio file, including missing information or inaccurately transcribed statements, which could lead to misinterpretation.

Considering that participants were requested to respond to the best of their knowledge to the same set of questions, were interviewed individually, and no time limit was placed on their responses, differences in reporting of themes/codes may potentially indicate a difference of opinion or knowledge about a subject. Thus, the frequency of codes from participants in each category for each theme was analyzed. Since the number of participants in each category differed, both the number and percentage of participants are indicated in the results.

Verbatim extracts are reported in the “government assistance program” and “accessibility for PVI” categories to provide representative views and opinions of groups of participants. Considering that many of the mentioned programs and services are not available outside of Peru, extracts were chosen to better understand the Peruvian context. Due to the number of participants and length of the interviews and transcripts, only a small sample of extracts is provided in the manuscript.

## 3. Results

### 3.1. Sociodemographic Characteristics of Participants

Twenty-nine participants aged 20 to 70 (19M (66%), 10F (34%)) were recruited from different areas of Peru. Three eye care professionals were ophthalmologists, while one was a specialist in rehabilitation. Educators interviewed were all working in the rehabilitation centers. Seven participants self-identified as having low vision with medical diagnoses, and twelve as functionally blind. Additional demographic information can be found in Table 1.

### 3.2. Perspectives on Assistive Devices

The use of assistive devices plays a crucial role in enhancing the independence of PVI. Participants responded to questions related to payment structures, access, and commonly used assistive devices; Table 2 highlights the top 3–4 codes from all groups. The findings reveal that the most frequently used devices among participants are white canes and smartphones, with 82% of PVI in Peru relying on these tools. White canes are typically acquired through donations, while smartphones are purchased by participants using their own resources. Notably, all blind participants reported using smartphones and computers equipped with applications such as JAWS, Envision, Lazarillo, and others. These tools enable effective communication and accessibility, as evidenced by their ability to seamlessly participate in Zoom interviews. Interestingly, only 9% of participants indicated that the government provides assistive devices, primarily tools for educational purposes such as braille slates, styluses, and abacuses used in schools. Eighty-three percent of rehabilitation and education professionals (R&E) mentioned that assistive devices such as the white cane are provided through donations. Medical professionals also mentioned that the white cane is the principal assistive device used by PVI and this device needs to be prescribed to get it from a government health practitioner. Additional information is shown in Table 2 below.

### 3.3. Perspectives on Vision Rehabilitation Services

Vision rehabilitation services (VRS) are among the most critical benefits that PVI should receive from the government. This right is enshrined in Law N° 29973 [10] and Supreme Decree N° 002-2014-MIMP [11]. Participants responded to questions related to the payment structure, access, and rehabilitation professionals in VRS. Our findings reveal that BP and MP identified the presence of two public VRS, such as the Social Health Program (ESSALUD) and the Centers for Basic Special Education (CEBES), as well as the private Rehabilitation Center for the Blind of Lima. LV identified the CEBES, while R&E identified only the CEBES and CERCIL. Regarding the access to VRS, both BP and R&E have highlighted the disparities between rural and urban areas due to the centralization of the VRS, the reliance on a personal support network, and the dependence on referrals to access those services. When we asked about the professionals in the VRS, both BP and R&E identified key components, including O&M training, braille education, computer skills instruction, and activities of daily living. However, despite these critical services, 83% of participants with low vision reported a lack of awareness about the availability of VRS.

Interestingly, MP emphasized the prevalence of non-functional rehabilitation services for patients with low vision, with 100% of respondents acknowledging this issue. Additional information is shown in Table 3 below.

### 3.4. Perspectives on Government Assistance Programs

Government assistance programs have a direct impact on PVI. Indeed, these programs could enhance social inclusion through education and promoting economic independence. Participants responded to questions regarding government assistance for work, education, transportation, and financial needs. The primary themes from these four important topics to improve the PVI quality of life are shown in Table 4 below.

#### 3.4.1. Employment

Our findings reveal that BP and R&E highlight the presence of two institutions established to support PVI in Peru: (1) The National Council for Disability (*Consejo Nacional de Discapacidad*, CONADIS) and (2) The Municipal Office for the Care of People with Disabilities (*Oficina Municipal de Atención a las Personas con Discapacidad*, OMAPED). However, out of seven LV respondents, only one was aware of CONADIS. Indeed, the majority of LV participants demonstrated limited knowledge about these institutions. In addition, BP, R&E and MP mentioned that within the government, there is a program aimed at assisting people with disabilities in the labor market. Despite this, 41% of respondents (specifically BP and R&E) criticized the inefficiency of the system in delivering effective support to people with disabilities, including PVI. Below are some stories shared by participants:R&E4: The problem is that no one enforces it, and no one complies with it. Programs aren’t implemented because, I believe, the people in charge of these offices don’t understand the work; they don’t know what it’s like to work with people with disabilities. Their only perspective—most of them, though not all—is that a person with a disability is someone who is there to receive a donation or support.BP2: What is the approach they take? Very well, here’s the job placement office. “Come, leave your résumé, and we’ll call you,” and it never happens. For them, that’s all, right? And no, no, they are not fulfilling the process of readaptation, real relocation, the four R’s, which is part of the International Labour Organization, Convention 159.BP8: The law says that 5% of the working population in a public institution must be a person with a disability and 3% of workers in a private institution that has more than 50 workers must be people with a disability. But it is not fulfilled either. Here in [Ayacucho], no public institution complies, for example, with this mandate that should be mandatory, they do not comply.LV6: I’ve heard about CONADIS because my sister worked there for a while, but… more, I don’t know. I only know that it’s the institution that helps people with disabilities.

#### 3.4.2. Education

Peru has ratified international agreements such as the UN Convention on the Rights of Persons with Disabilities, which emphasizes inclusive education. Our study shows that 50% of all participants recognize that inclusive education is being implemented by the government programs, but the transition is challenging. Thus, 50% of all participants reveal that there is a lack of programs supporting education, and 27% mention that the lack of teacher preparation is the main issue during this transition. Our results also indicate that LV and MP are the groups with a lack of knowledge about the government assistance programs aimed at inclusive education. Notably, our results also show that only R&E know about the “Support Service and Advice for the attention of Special Educational needs” program, which was established to support and accompany the PVI in the inclusive schools. Here are some arguments mentioned by the participants:BP8: For example, allocating a budget for educational institutions to be accessible to people with disabilities, including people with visual disabilities. When a parent takes a visually impaired child to school to register, they are rejected. Why? Because many times it is not because they are bad, but because of the teacher. They are afraid of assuming that responsibility because they are not prepared.R&E3: …a specific law for the inclusion of children with disabilities in regular schools. There is a budget, which is budget (106) of the Ministry of Education. This is supposed to provide the supplies such as cane devices, the strips, braille machines, but that is not coming, this is still not being implemented in an organized or coordinated manner.R&E4: Now [PVI] have participation or are physically included in the schools in their area, but I consider that the development and skills are not being given to them in regular schools because there is no material and there are not yet enough prepared teachers.

#### 3.4.3. Transport

Transport for PVI in Peru is also supported by the law. Indeed, the law 29973 mandates that public transportation systems must be accessible to all individuals, including those with VI [10]. Seventy-five percent of all participants know the benefit of the free public transportation for PVI. Notably, 75% and 55% mentioned that there is non-compliance with the regulations, and there is a lack of respect from drivers. Below are some comments from participants:BP1: It is said that we have the benefit of, for example, not being charged for the fare when we take public transport. However, there are certain drivers who do not respect this regulation and demand that we pay for it even if we have a card issued by CONADIS.BP4: Oh? You don’t have fare? Then just get off. There’s a sense of disdain. If you can’t pay, it’s that extreme. It’s not accepted by the transport operators or fare collectors because they lack education. And basically, they don’t have a family member with a disability. Because if you don’t have a family member with a disability, you won’t understand. Why? Because that’s just how the culture is, unfortunately.LV3: Well, there are many people who, when you show your yellow card to the collectors, the driver gives you a dirty look, they want to get you off, they even hit you. Now, every time I get on public transport, I pay.R&E1: For example, when you have the yellow card, you are entitled to free passage, but it is not fulfilled. And, for example, the user gets on a bus, and you show them your card, they do not pick you up on the next ride.R&E2: It’s a question of awareness. There are some people who are more aware than others. We cannot generalize, because there are some who do respect the yellow card and there are others who do not, they do not give the opportunity.

#### 3.4.4. Financial Needs

Although Peru is still considered a low-income country, the government provides financial assistance to people with disabilities, including individuals with visual impairments, through the CONTIGO program. Seventy-seven percent of participants revealed that they are aware of the financial assistance offered by the CONTIGO program. Among them, 54% identified extreme poverty and severe disability as the primary eligibility criteria for accessing this benefit, and 15% mentioned that there is no financial aid for PVI. Interestingly, none of the LV or BP mentioned receiving financial aid from the government.

### 3.5. Perspectives on Accessibility for People with Visual Impairments

Accessibility for PVI is undoubtedly a critical component of Peru’s efforts toward greater inclusivity. In this study, we explored the barriers faced by PVI in accessing education and achieving independence both indoors and outdoors. The results reveal that BP identify education as a significant challenge, primarily due to inadequate teacher training (45%). Additionally, social stigmatization (27%) and unequal educational opportunities (27%) are seen as major barriers to accessing education. Similarly, LV and R&E also highlighted the lack of teacher preparation as a key obstacle (67%). In contrast, MP pointed to economic limitations as the primary barrier preventing PVI from accessing education (100%). Orientation and mobility are recognized as essential yet challenging skills for PVI to attain independence. When discussing barriers to indoor and outdoor navigation, the findings demonstrate considerable alignment among all four groups surveyed. Eighty-three percent of BP revealed dependence on others, 75% mentioned the presence of street vendors, and 58% said that there is an absence of inclusive areas. Similarly, 86% of LV respondents also mentioned dependence on others, while 42% declared a lack of social awareness. MP also revealed that non-empathetic pedestrians and a lack of social awareness are the most prominent barriers that PVI face while they are indoors or outdoors. Here are some quotes mentioned:BP6: It is the fear of teaching a person with visual impairment, because they do not have the resources or simply because they do not want to do it.BP11: Teachers should be prepared to care for young people with disabilities. There are many difficulties in this regard, I was even asked: how am I going to evaluate you? So, higher level teachers must be strengthened in this area, so that they are prepared.LV1: The university does not have a system for her [visually impaired person] and they have verbally asked her to leave, to find another option.R&E4: Teachers have not assumed this responsibility. This is because the authorities at universities or institutes have not become aware of this responsibility.BP4: It is believed that visually impaired people are not productive. I mean, why am I going to hire someone? How are they going to get around, how are they going to do it? No, I’d rather hire someone who can see.” And that also happens in the family “poor little girl, I have to take care of the poor thing” and they don’t let her do anything.BP12: The issue of bullying is terrible. I have two or three friends who stopped studying because the bullying was terrible.LV6: They are not empathetic at all. I even stopped having friends.MP1: Health systems are imposing the use of elevators, ramps. In my opinion it is quite important, yes. But exclusively for a patient with visual impairment… There is none.BP3: They [transport services] should have staff who are willing to help when they see a person with visual disability.BP8: For example, in some streets they put the motorcycle in the middle of the sidewalk, or the car has a tire pushed up onto the sidewalk. I don’t know why they do that. You go with your white cane, and you crash into that. It’s a lack of sensitivity from the part of the population itself.R&E2: Raising awareness and training drivers in all respects, with more stringent regulations or so that they also feel that it will affect them in some way if they do not comply.

Notably, all groups emphasized the importance of public education as a critical factor in supporting daily living activities and fostering greater independence for PVI. The primary themes regarding accessibility are shown in Table 5.

### 3.6. Perspectives on Eye Care Services

PVI face significant challenges accessing eye care services due to several reasons, such as geographic and economic barriers [19]. All participants revealed that ophthalmologists are the professionals in charge of eye care. LV (29%) and R&E (33%) also mentioned the presence of optometrists as professionals involved in the eye care services. Interestingly, MP (67%) mentioned the presence of low vision professionals who are taking care of non-functional rehabilitation.

Regarding the payment structure and access to eye care services, all participants acknowledged the availability of eye care services across various healthcare providers, including hospitals under the Ministry of Health (such as the National and Regional Institute de Ophthalmology), ESSALUD facilities, and private clinics. However, access to these services varied in terms of administrative processes and geographic distribution.

Additionally, 42% of BP respondents, 83% of LV respondents, and 50% of R&E respondents noted that scheduling an appointment was necessary to consult an ophthalmologist. Of interest, 25% of BP respondents, 17% of LV respondents, and 17% of R&E respondents highlighted that specialized eye care services were predominantly concentrated in Lima, the country’s capital. The primary themes regarding eye care services are shown in Table 6.

### 3.7. Differences and Similarities Between LV and BP Perspectives

Since BP and LV are the main users of the VRS and the government assistance programs, we investigated the differences and similarities between those groups. Using the two-case model from MAXQDA, our results show substantial differences and similarities between BP and LVP. As seen in Figure 1, our results indicate that, unlike BP, interviewed LV reported very little knowledge about the VRS and the government assistance programs. Of interest, several perspectives (codes) often overlap; however, for eye care services, there are no substantial differences.

## 4. Discussion

This qualitative research explored different perspectives about living with visual impairment in Peru. Interviewing four different groups allowed us to identify differences regarding the challenges faced by the PVI. Interestingly, blind participants and those with low vision have different perspectives about the programs from the government and the accessibility for PVI.

### 4.1. Assistive Devices

The emergence of new technologies has shown significant potential to improve the quality of life for PVI [23,24,25]. Our study shows that the white cane is an important tool in orientation and mobility with high prevalence of use among BP; however, LV are reluctant to use it, which is likely due to the stigma leading to embarrassment and concern about being stared at [26]. On the other hand, our findings reveal that 100% of BP reported using smartphones and applications, highlighting their positive impact on activities of daily life. Indeed, smartphones offer the crucial advantages of accessibility and versatility compared to dedicated assistive devices [27]; however, the use of these tools remains largely confined to applications related to reading and orientation. Furthermore, the accessibility of assistive technologies is hindered by their high costs, further limiting widespread adoption [28,29]. Recently, the Global Education Monitoring Report team from UNESCO mentioned that in Latin America and the Caribbean, including Peru, the assistive technology is often not available due to a lack of resources or a lack of training [30], which restricts the ability of PVI to effectively integrate these tools into their daily routines. While assistive devices continue to evolve, offering numerous benefits that could significantly impact the quality of life [28,31], persistent economic, social, and infrastructural barriers must be addressed to ensure equitable access and utilization.

### 4.2. Vision Rehabilitation Services

Like in other countries, vision rehabilitation services in Peru are aimed at improving the quality of life of PVI, and these services include training in the use of devices and technology, training in O&M skills, support in daily life activities, etc. [32]. Those interviewed mentioned that these services are provided mainly by the Social Health Insurance (ESSALUD), CERCIL (Rehabilitation Center for the Blind), and the Ministry of Education through the Special Basic Education Centers (CREBES). CERCIL is the center that offers the most complete rehabilitation program, yet it is private and only available in Lima. The Ministry of Health and ESSALUD do not offer functional rehabilitation, and this task is achieved by the Ministry of Education; however, there are challenges to reaching PVI, particularly in rural and underserved areas. Our study also indicates that access to the VRS is through referrals. However, more studies are needed in order to understand how referrals influence services provided to PVI. Interestingly, similar to studies reported from North America and the United Kingdom, where VRS are more available, our results indicate that people with low vision reported a lack of awareness of VRS and refusal to use the VRS due to a perceived lack of need or social considerations, among others [27,33]

Our participants report that CERCIL has professionals trained to address the diverse needs of PVI. These professionals in the government receive experience-based training but are not certified or specialists. Professionals, as well as rehabilitation services, are needed to expand across the country to ensure equitable distribution of services. Collaboration between government agencies, educational institutions, and NGOs is essential to strengthening the field of visual rehabilitation in Peru.

### 4.3. Government Assistance Programs

The Peruvian government has implemented several assistance programs and policies to support PVI, aiming to improve their quality of life, promote inclusive education, and ensure access to essential services [10,34]. These programs are primarily managed by the Ministry of Health (MINSA), the Ministry of Education (MINEDU), and the Ministry of Development and Social Inclusion (MIDIS). Our results indicate that people with blindness and low vision know that these assistance programs promote job opportunities; however, they mention that these programs are still inefficient. Similar results were obtained regarding available programs for transportation, education, and financial assistance. Unfortunately, there are no studies or data showing all the challenges that contribute to their inefficiency. Several likely reasons impede a good outcome of these programs, including limited resources and gaps in implementation (bureaucracy), centralization of the programs, lack of specialized training and professionals, stigma and social exclusion, as well as inadequate monitoring and evaluation. Applying a systematic approach to addressing these challenges, the Peruvian government may improve the effectiveness of the programs. Collaborations with NGOs and private organizations could further promote the achievement of these goals.

### 4.4. Accessibility for the Visually Impaired

Accessibility for PVI is part of the General Law for Persons with Disabilities [10]. This law mandates equal opportunities and accessibility for PVI. In addition, there is a National Accessibility Plan 2018–2025 [35]. Our study shows that a lack of teacher preparation is the main barrier to accessing an inclusive education according to the BP, LV, and R&E. Similar results were reported by the Ministry of Education (2016) [36], the Ombudsman [37], and Valdivieso (2020) [38]. Research to improve inclusive education is almost nonexistent in Peru, despite that working with the school community to improve school climate for inclusive education had positive effects on the inclusion [38]. Interestingly, in 2006, the Ministry of Education created the “Support Service and Advice for the attention of Special Educational Needs” (SAANEE) program to promote inclusive education for students with disabilities, including those with visual impairments. SAANEE operates within the framework of Peru’s inclusive education policy and provides specialized support to students, teachers, and families [39]. Thus, it has been reported that in 2015, about 1940 teachers received virtual training courses on the teaching and learning process for children with visual disabilities [28]. Nevertheless, the implementation of inclusive education is still challenging [38]. Indeed, our results indicate that the lack of teacher preparation and support programs, social stigmatization, and inequalities of educational opportunities for PVI are among the most frequent codes reported by BP, R&E, and MP. We speculate that increased funding and greater awareness of its services, as well as providing adequate monitoring of the indicators on inclusive education, would enhance its adoption [37]. Moreover, we agree with the report by the World Bank about teachers needing structural support, including better salaries and working conditions, especially in rural areas [28].

Indoor and outdoor mobility are very critical for PVI, with several impacts, including physical, social, and mental health [29]. Independent mobility also has a tremendous impact on social interactions and performing daily activities [40,41]. Our results indicate that BP, LV, and R&E report the necessity or dependence on others to navigate safely both indoors and outdoors. Indeed, several accessibility features should be modified to promote independence of PVI, such as additional braille signage, incorporation of tactile markings, improving sidewalk conditions, location of street vendors, and, most importantly, improving social awareness. In Peru, physical accessibility for PVI, both indoors and outdoors, remains a significant challenge despite legal frameworks and efforts to promote inclusion. Participants in our study suggest that accessibility challenges include compliance with the legislation, inclusive design, and structural changes.

Interestingly, when we asked the participants for suggestions to improve the accessibility of PVI, all groups highlighted sensitization (education of the society). Indeed, attitudinal barriers have a great influence on personal development and are associated with stigma, discrimination, and mistreatment, which are considered more harmful than physical barriers [42]. Previous studies have reported that PVI who have experienced discrimination were more likely to report depression and poor quality of life [43], with negative impacts on social interactions [44]. Based on the results presented in our study as well as previous studies, it is imperative to promote strategies to improve awareness and educate the general public to reduce the discrimination and negative perceptions related to PVI.

### 4.5. Eye Care Services

Our results indicate that all groups acknowledge the existence of public and private institutions providing eye care services. However, PVI face significant challenges in accessing these services, particularly those residing in rural areas, as highlighted by BP, LV, and R&E. In fact, it has been reported that the socioeconomic status of many families in the rural areas poses a major barrier, severely limiting their access to healthcare services [19]. In this context, creating blindness prevention programs targeting people with low socioeconomic status in the Peruvian Amazon has been suggested [45]. Another point that we would like to highlight is the role of optometrists in Peru. According to our interviews, unlike in developing countries, optometrists are still facing some challenges, such as limited training opportunities and a lack of public awareness about their profession.

### 4.6. Limitations

Though the participants are from different cities across the country, it is likely that most of the responses do not represent the reality in different communities, specifically in rural areas. Over the course of this study, we interviewed persons from different groups, and we encouraged a diversity of responses. In this context, the views and opinions expressed do not necessarily reflect those of all individuals in each category.

## 5. Conclusions

Visual disability is the most frequent disability in Peru, and investing in eye health services could have positive benefits in the economic development [46]. Our results reveal that multiple systemic barriers severely hinder the inclusion of PVI into Peruvian society. In fact, many of the barriers highlighted are principally societal and touch on multiple aspects of the everyday life of PVI, including the use of assistive devices, education, employment, and transportation. Though medical professionals are aware of some of the realities of PVI, their knowledge about many of these subjects is more limited than that of the other groups interviewed. Individuals with lived experience, both those with low vision and blindness, should be consulted, along with rehabilitation professionals and educators, to ensure that investments to improve the lives of PVI are well targeted. A targeted approach to minimizing their impact on the everyday lives of PVI would create more educational and economic opportunities for this vulnerable population. Although this study gives an overview of living with a visual impairment in Peru, further research should be conducted in each of these areas to examine the implementation of services to improve the lives of PVI and promote their full potential.

## Figures and Tables

**Figure 1 ijerph-22-00984-f001:**
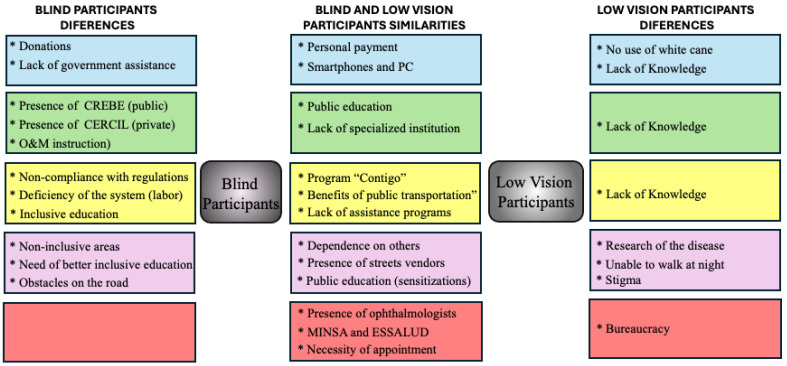
The two-case model. Differences between BP and LV coding report from top to bottom. Blue: assistive devices; green: VRS; yellow: government assistance programs; pink: accessibility; red: eye care services.

**Table 1 ijerph-22-00984-t001:** Sociodemographic information about participants.

Group	ID	Age	Gender	Residence	Occupation	Visual Impairment
Blind	BP1	20	F	Trujillo	Undergraduate student	Viral infection
participants	BP2	49	M	Lima	Technician and law student	Retinal detachment
	BP3	52	F	Piura	Technical diploma	Congenital glaucoma
	BP4	36	M	Ayacucho	Musician and undergraduate studies	Chronic uveitis and glaucoma
	BP5	36	M	Trujillo	Physiotherapist	Congenital cataract, glaucoma
	BP6	33	F	Ayacucho	Musician	Retinitis pigmentosa
	BP7	68	M	Arequipa	Professor (retired)	Optic atrophy
	BP8	59	M	Ayacucho	Undergraduate studies	Retinitis pigmentosa
	BP9	39	M	Lima	Sociologist and law student	Congenital glaucoma
	BP10	52	M	Trujillo	Physiotherapist	
	BP11	44	M	Juliaca	Social communicator	Optic nerve damage
	BP12	47	M	Chimbote	Technician	Optic nerve damage
Low vision	LV1	40	M	Tingo Maria	Kitchen helper	Retinitis pigmentosa
	LV2	41	F	Piura	Elementary teacher	Retinitis pigmentosa
	LV3	60	M	Piura	Electrician	Retinal detachment (OD)
	LV4	42	M	Pucallpa	Soft beverage salesperson	Retinitis pigmentosa
	LV5	53	M	Barranca	Piano teacher	Retinitis pigmentosa
	LV6	36	F	Ayacucho	Housewife	Usher’s syndrome
	LV7	75	F	Trujillo	Housewife	AMD, cataract
Educator	R&E1	57	M	Arequipa	Special education teacher	NA
	R&E2	47	F	Trujillo	High school professor	Retinitis pigmentosa
	R&E3	56	M	Arequipa	Education and administration	Retinitis pigmentosa
	R&E4	48	M	Arequipa	Education	Congenital glaucoma
	R&E5	57	F	Ayacucho	Physical education	NA
	R&E6	59	F	Chincha	Pedagogical technician	NA
Medical professionals	MP1 MP2	55 51	M F	Trujillo Lima	MD, ophthalmologist MD, rehabilitation	NA NA
	MP3	54	M	Lima	MD, ophthalmologist	NA
	MP4	55	M	Lima	MD, ophthalmologist	NA

M: Male; F: Female, MD: Medical doctor; BP: Blind participants; LV: Low vision participants; R&E: Rehabilitation practitioners/educators; MP: Medical professionals; AMD: Age-related macular degeneration; NA: not applicable.

**Table 2 ijerph-22-00984-t002:** Use of assistive devices.

Thematic	Codes	BP	LV	R&E	MP
		Frequency (%)			
Payment structure	Free materials (insurance)				1 (25)
	Free apps	6 (50)		3 (50)	1 (25)
	Free materials (government)	2 (17)			
	Personal payment	12 (100)	1 (14)	5 (83)	2 (50)
Access	Prescription			1 (17)	1 (25)
	Importation			1 (17)	1 (25)
	Lack of knowledge (access)		1 (14)		
	Social network or friends	5 (42)	1 (14)	1 (17)	1 (25)
	Lack of government assistance	3 (25)		3 (50)	1 (33)
	Bureaucracy	2 (17)	1 (14)	1 (17)	
	Donations	7 (58)		5 (83)	
Devices	Minimal use of the white cane		5 (71)		
	Stylus	6 (50)		3 (50)	
	Writing slate	5 (42)		4 (67)	
	White cane	12 (100)	1 (14)	6 (100)	4 (100)
	Computers	7 (58)	3 (43)	2 (33.3)	1 (25)
	Smartphones	12 (100)	3 (43)	6 (100)	2 (50)

BP: Blind participants; LV: Low vision participants; R&E: Rehabilitation practitioners/educators; MP: Medical professionals.

**Table 3 ijerph-22-00984-t003:** Vision rehabilitation services.

Thematic	Codes	BP	LV	R&E	MP
		Frequency (%)			
Payment structure	ESSALUD-CERP—free	9 (75)			1 (25)
	CEBES—free education	4 (33)	1 (14)	4 (67)	1 (25)
	CERCIL—private	1 (8)		3 (50)	1 (25)
Access	Rural–urban divide	4 (33)		2 (33)	
	Personal support networks	3 (25)		3 (50)	
	Referrals	7 (58)	1 (14)	4 (67)	2 (50)
Professionals	Low vision department—absence of functional rehabilitation			1 (16.7)	4 (100)
	Lack of knowledge		6 (86)		
	Lack of specialized institutions	3 (25)	1 (14)	3 (50)	
	CREBES practitioners	4 (33)		2 (33)	
	Activities of daily living	2 (17)		5 (83)	
	O&M	7 (58)		5 (83)	
	Computer lessons	4 (33)		4 (67)	
	Braille education	5 (42)		4 (67)	

BP: Blind participants; LV: Low vision participants; R&E: Rehabilitation practitioners/educators; MP: Medical professionals.

**Table 4 ijerph-22-00984-t004:** Government assistance programs.

Thematic	Codes	BP	LV	R&E	MP
		Frequency (%)			
Work	Lack of knowledge		1 (14)	1 (17)	2 (50)
	CONADIS assists people with disabilities	2 (17)	1 (14)	3 (50)	
	Inclusion of people with disabilities in the labor market	6 (50)		5 (83)	1 (25)
	Inefficiency of the system	7 (58)		2 (33)	
	OMAPED	5 (42)		1 (17)	
Education	Inequities in education			1 (17)	1 (25)
	SAANEE support and accompaniment			4 (67)	
	Lack of knowledge		3 (43)		1 (25)
	Inclusive education	6 (50)		4 (67)	1 (25)
	Lack of teacher preparation	3 (25)		3 (50)	
	Lack of support programs	6 (50)	1 (14)	3 (50)	1 (25)
Transport	Centralized	1 (8)	2 (29)	1 (17)	
	Non-compliance with regulations	9 (75)		6 (100)	
	Benefits of public transport	9 (75)	1 (14)	5 (83)	
	Lack of respect from drivers	7 (58)	1 (14)	3 (50)	
Financial	Little knowledge	1 (8)	5 (71)		1 (25)
needs	Assistance for extreme poverty	6 (50)	1 (14)	3 (50)	1 (25)
	No financial aid	2 (17)	1 (14)	1 (17)	
	Program “CONTIGO”	9 (75)	3 (43)	4 (67)	1 (25)

BP: Blind participants; LV: Low vision participants; R&E: Rehabilitation practitioners/educators; MP: Medical professionals.

**Table 5 ijerph-22-00984-t005:** Accessibility for the visually impaired.

Thematic	Codes	BP	LV	R&E	MP
		Frequency (%)			
Education	Inequality of educational opportunities	3 (25)	2 (29)	1 (17)	
	Social stigmatization	3 (25)	1 (14)	3 (50)	
	Economic barriers	2 (17)		1 (17)	1 (25)
	Lack of teacher preparation	5 (42)	2 (29)	4 (67)	
Indoors and	Lack of support staff			2 (33)	
Outdoors	Centralism and marginalization				1 (25)
	Obstacles on the road	4 (33)		4 (67)	
	Deteriorated or inadequate sidewalks	6 (50)	1 (14)	4 (67)	
	Lack of social awareness	6 (50)	3 (43)	2 (33)	2 (50)
	Dependence on others	10 (83)	6 (86)	3 (50)	
	Presence of street vendors	9 (75)	1 (14)	5 (83)	
	No tactile markings	4 (33)	1 (14)	2 (33)	2 (50)
	Non-inclusive areas	7 (58)		4 (67)	1 (25)
	Lack of elevator announcements	2 (17)	1 (14)	2 (33)	1 (25)
	Non-empathetic pedestrians	9 (75)	3 (43)	1 (17)	2 (50)
Suggestions	Compliance with legislation	4 (33)	1 (14)	1 (17)	
	Public education (sensitization)	7 (58)	4 (57)	5 (83)	2 (50)
	Training of public officials	3 (25)	2 (29)	4 (67)	2 (50)
	Structural change	3 (25)	1 (14)	2 (33)	
	Inclusive design (architecture)	6 (50)	2 (29)	3 (50)	1 (25)
	Street signage	3 (25)	2 (29)		

BP: Blind participants; LV: Low vision participants; R&E: Rehabilitation practitioners/educators; MP: Medical professionals.

**Table 6 ijerph-22-00984-t006:** Eye care services.

Thematic	Codes	BP	LV	R&E	MP
		Frequency (%)			
Payment	ESSALUD	2 (17)	3 (43)	2 (33)	1 (25)
structure	MINSA (INO, IRO)	8 (67)	3 (43)	5 (83)	2 (50)
	Personal payment (private clinics)	6 (50)	5 (71)	3 (50)	2 (50)
Access	By appointment	5 (42)	5 (71)	3 (50)	
	Geographic gap (centralized)	3 (25)	1 (14)	1 (17)	
Professionals	Low vision				2 (50)
	Ophthalmologist	12 (100)	7 (100)	6 (100)	4 (100)
	Optometrists	3 (25)	2 (29)	2 (33)	

BP: Blind participants; LV: Low vision participants; R&E: Rehabilitation practitioners/educators; MP: Medical professionals.

## Data Availability

The data generated during and/or analyzed during the current study are not publicly available. These data include participant identifying information that is sensitive and confidential, but it is available from the corresponding authors upon a reasonable request.

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
