# Peer review of "The Experiences of Living with a Visual Impairment in Peru: Personal, Medical, and Educational Perspectives"

_ijerph, 2025, doi:10.3390/ijerph22070984_

Round 1

Reviewer 1 Report

Comments and Suggestions for Authors

Comments to Authors

Thank you for this original article, which covers an important subject and is a pleasure to read. The methodology is generally correct. A few points could be explored further to improve the quality of the manuscript.

In general, inclusive language should be used. In a few places in the manuscript (for example, L23, L172, L324), we find “the visually impaired”. It would be better to write, people/person/individual with/living with visual impairment.

For references in the text, “et al” comes from Latin and should therefore be written in italics.

L123: the study explored

L139: the overall goal of the study was

L152: theoretical data saturation

L152-153: Several types of participants were recruited but it is not clear whether theoretical data saturation was obtained for each type of participant or for all participants. This should be clarified.

L157: visual impairment: when an abbreviation is defined, it must be used throughout the text.

In line 170, the authors state that a member of the team conducted the interviews. Line 178 reads “the authors”. How many researchers actually conducted the interviews?

L180: Were the transcripts verbatim?

Was the analysis inductive, deductive or abductive?

Was a rigorous data processing method used (e.g. thematic analysis)?

L182-183: Only one researcher carried out the analyses. How did the authors ensure the reliability of their analyses?

What strategies were used to ensure the quality of the research?

 How were the data reported? The results include frequencies (quantitative) and text extracts (qualitative). These choices should be justified, and more details given about the methodology, the calculation of frequencies and the choice of extracts.

L187: write percentages of men and women.

Table 1: define abbreviations in the legend.

L244 & L247: define the abbreviations in the text for the first utilization.

L295: do not begin a sentence with a number.

Figure 1: Figure 1 is cut.

Reviewer 2 Report

Comments and Suggestions for Authors

The abstract gave a good overview of the study in context.

The introduction gave a detailed background of the study

The objectives and research questions were well spelt out

I enjoyed reading the results and the discussion.

This article will be a good referral point for scholars in Peru and the world at large

Kindly take note of these few recommendations:

Don't you think the title could be rewritten as:

The Experiences of Living with a Visual Impairment in Peru: Personal, Medical, 2 and Educational Perspectives (think about it)

In line 265 on page 8 there was a mention of name of a place “Ayacucho” if it is a pseudonym indicate it in brackets.

It would be very good if you indent all your direct quotes from the participants

Under the discussion line 407, I feel the sentence “This qualitative research explored different perspectives about how is living with visual impairment in Peru” should read [This qualitative research explored different perspectives about living with visual impairment in Peru].

Under vision rehabilitation services on line 440, the word “undeserved” should be [underserved]

Under the accessibility for visually impaired, the two sentences on line 470 “Accessibility for PVI is part of the General Law for Persons with Disabilities. This law mandates equal opportunities and accessibility for PVI” will need a reference to support those sentences.

Check the sentence on lines 506 and 507 “… which are considered more harmful that physical barriers” Is the word “that” Not supposed to be [than]?

Check the sentence on line 509 and 510 “Based on our results and previous studies is imperative to promote …” Not very clear

Sentence on line 549. “All author reviewed …” The word author should be [authors]

Comments on the Quality of English Language

The quality of English Language was very good. Just few sentences need to be crosschecked
